

# Spatial distribution and post-depositional diffusion of stable water isotopes in East Antarctica

Mahalinganathan Kanthanathan[1], Thamban Meloth[1], Tariq Ejaz[1], Bhikaji L Redkar[1], and Laluraj C Madhavanpillai[1]

[1]National Centre for Polar and Ocean Research, Headland Sada, Vasco-da-Gama, Goa – 403804

**Correspondence:** K. Mahalinganathan (maha@ncpor.res.in)

**Abstract.** We have analysed the spatial variations in the mean stable water isotopic values, snow accumulation patterns and moisture sources along coast to inland transects in central Dronning Maud Land (cDML) and Princess Elizabeth Land (PEL) regions of East Antarctica. The $\delta$D and $\delta^{18}$O varied systematically from coastal to inland regions in cDML and PEL regions in response to the surface air temperature. While the elevation effect was not clearly visible, the isotope variations appeared
to be associated with snow accumulation in cDML region and temperature in PEL region, which ultimately are associated with elevation. Further, a clear influence of topography on the snow accumulation was observed in cDML region. Such an observation was not recorded in PEL transect, apparently due to the strong snow redistribution in this region due to katabatic winds. The moisture sources to the study areas were identified using HYSPLIT backtrajectory calculations. The major sources of precipitation during summer arrived from the south Atlantic ocean in the cDML and the Indian Ocean in PEL. During winter,
the sources of precipitation in cDML extended upto Weddell Sea while in PEL, the sources extended upto 50°S in the Indian Ocean. In order to understand the post-depositional isotope diffusion processes in firn, a firn core which was drilled close to the cDML transect, five years after the snow core transect, was analysed in comparison with snow records. Our study showed a significant isotope amplitude diffusion with a diffusion length of 6 cm from the surface to 4 m depth in 5 years.

## 1 Introduction

Ever since Dansgaard (1964) successfully explained the dependency of stable isotopic composition of precipitation on physical
parameters such as temperature, latitude and altitude, the stable water isotopes ($\delta$D and $\delta^{18}$O) have become the primary proxy tool in paleoclimate studies. As a result, the high resolution $\delta$D and $\delta^{18}$O records from the polar regions have been continuously used to interpret the temperature and snow accumulation information from decadal to millennial time scales (e.g. EPICA, 2006;
2k PMIP3 group, 2015). However, the stable water isotope records in Antarctic snow depends on multiple factors such as the temperature of the moisture source region where evaporation occur, followed by fractionation due to subsequent condensation and evaporation cycles during transport and finally the temperature at which the precipitation forms (Landais et al., 2017).





Though ice core $\delta$D and $\delta^{18}$O records are often assumed to represent climatic signals of an entire region, spatial variation of these isotope records indicate complex signals that are often not straightforward to interpret. Factors such as accumulation rates,

erosion and redistribution by wind contributes to such variations in spatial scale (Fisher et al., 1985; Town et al., 2008). Another important factor that could influence the isotope based climate records in Antarctic ice is the role of extreme precipitation events (Turner et al., 2019). The deuterium excess (d), which is a second-order parameter defined by combination of $\delta$D and $\delta^{18}$O (d $= \delta$D $- 8.\delta^{18}$O), reflects the behaviour of these two isotopic species occurring during the kinetic and equilibrium fractionation (Jouzel and Merlivat, 1984). The association between 'd' and the climatic parameters such as the surface temperature, relative

humidity and the wind speed in the moisture source region have been established in previous studies (Jouzel et al., 1982; Jouzel and Merlivat, 1984; Petit et al., 1991).

Paleoclimate studies based on ice cores primarily rely on stable water isotope records to provide paleoclimate information. However, these stable isotope records are modified post deposition by diffusion processes (Johnsen, 1977; Whillans and Grootes, 1985). Even though the stable isotope composition in top layers of snow and firn is influenced by diffusion of water

vapour and snow densification (Cuffey and Steig, 1998; Hörhold et al., 2011), no net change in isotopic composition were observed due to these processes.The diffusion rate in ice beneath the firn is negligible and therefore gets preserved for much longer time after surviving the diffusion in the top layers (Johnsen, 1977). Studies by Steen-Larsen et al. (2014) and Ritter et al. (2016) have shown isotopic exchanges in the boundary layer on daily scale and on diurnal scales in the polar regions. The isotope signals on the near-surface snow and firn undergo fractionation due to sublimation and condensation processes as

observed both in-situ and laboratory studies (Stichler et al., 2001; Sokratov and Golubev, 2009). Apart from the temperature-based fractionation, other factors such as relative humidity and wind speed also influence the post-depositional variations in the ice core isotope signals.

The climate records in ice cores reflect the conditions at which snowfall occurred. These records are influenced by various parameters such as the season at which the precipitation occur and variations in sources of moisture for precipitation. In order

to better interpret the ice core stable isotope records, it is important to understand their present variability and interrelationships using modern precipitation and surface snow. The present study analyses the spatial variability of stable water isotopes in terms of latitudinal and altitudinal effects, variation in snow accumulation rates and possible moisture sources in two geographically distinct regions of East Antarctica – Princess Elizabeth Land (PEL) and central Dronning Maud Land (cDML). We also discuss the effect of diffusion of stable isotope signals in the top layers of snow by comparing with the data from an ice core that was

drilled close to the transect in the cDML region.

## 2  Materials and Methods

The study regions are in two different sectors of East Antarctica – the Princess Elizabeth Land (PEL) and the central Dronning Maud Land (cDML), facing the Indian Ocean and the Atlantic Ocean sectors, respectively (Fig. 1). Majority of snow cores (41) were recovered from PEL and cDML transects during the same austral summer of 2008–09 and five more cores were recovered

from the inland section of cDML during 2009–10. The transects were categorized into coastal, mountainous and inland regions



based on topography as mentioned in detail in our earlier publications (Mahalinganathan et al., 2012; Mahalinganathan and Thamban, 2016). The snow cores were collected using KOVACS Mark IV system and each core was typically 1 meter long and 14 cm diameter. After recovery, the snow cores were transferred directly into pre-cleaned high-density polyethylene bags and sealed immediately to avoid contamination during storage and shipped under frozen conditions to the National Centre for Polar and Ocean Research, India. The cores were stored at –20°C till analysis. The snow cores were sub-sampled at a 5 cm resolution under a clean laminar hood. A total of 667 sub-samples were processed where the outer layers were removed manually by a clean ceramic knife and the innermost cube was utilized for density measurements from which the snow accumulation rate values were calculated (Mahalinganathan et al., 2012). A portion of each sub-sample was transferred to a 10 mL teflon vials and sealed for further stable isotope analyses.

All sub-samples were melted and immediately analysed for $\delta$D and $\delta^{18}$O values in the Ice Core Laboratory of National Centre for Polar and Ocean Research using a dual inlet, Isoprime Isotope Ratio Mass Spectrometer, following standard procedures (Naik et al., 2010) which had an external precision of 0.05 ‰ . Fresh samples were also analysed using a unique laser-based Off-Axis – Integrated Cavity Output Spectroscopy (OA–ICOS) Triple Isotope Water Analyser (TIWA) by Los Gatos Research. The melted samples were introduced directly into the TIWA using a Hamilton 1.2 $\mu$L zero dead volume syringe via an auto injector equipped with a heated ( 85°C) injector block. In order to eliminate inter sample memory effect, four preparatory injections (injections which are not measured), followed by five measurement injections were run for each sample. The last five injections were averaged to produce a single, high-throughput sample measurement. The analytical precision of measurements for $\delta$D was ±0.5‰ and $\delta^{18}$O was ±0.1‰ using the TIWA system. Details of sampling locations and analytical results are provided in tables 1 and 2 for cDML and PEL, respectively.

The seasonality in snow cores was determined by establishing the summer and winter peaks in isotope records where a minimum amplitude of 4‰ between summer and winter was used to differentiate these peaks as detailed in figure 2 our previous publications (Mahalinganathan et al., 2012; Mahalinganathan and Thamban, 2016). Near-surface temperature measurements were estimated for the year 2008 from Regional Atmospheric Climate Model version 2.3 (RACMO 2.3) output with a horizontal resolution of 27 km (Wessem et al., 2014). Isotope results from an ice core (IND33/B8, ~101 m) drilled close to the cDML transect was used in order to estimate the diffusion of stable water isotopes in firn. The firn diffusion model originally described in Johnsen et al. (2000) was employed to estimate the diffusion length using the equations provided in Gkinis et al. (2014), Dee et al. (2015) and Münch and Laepple (2018). These calculations were performed using the measurements from the ice core site (site specific parameters) with an average firn density of 340 $\mathrm{kg\,m^{-3}}$, a mean surface temperature of –22°C, local mean surface pressure of 800 mbar and a local accumulation rate of 300 $\mathrm{kg\,m^{-2}yr^{-1}}$. The density and local accumulation rates were directly measured from the snow cores and ice core data while the temperature and local mean surface pressure values were derived from the RACMO output.





## 3  Results

Snow accumulation rates at both cDML and PEL regions were generally high (averaged values) and reduced from coastal to inland sections as expected (Tables 1 and 2). Accumulation in the coastal section of cDML ranged between 313–410 $\mathrm{kg\,m^{-2}yr^{-1}}$

(mean 354 $\mathrm{kg\,m^{-2}yr^{-1}}$) while in the inland section, it ranged between 104–250 $\mathrm{kg\,m^{-2}yr^{-1}}$ (mean 154 $\mathrm{kg\,m^{-2}yr^{-1}}$) with the mountainous section ranging between 200–320 $\mathrm{kg\,m^{-2}yr^{-1}}$ (mean 235 $\mathrm{kg\,m^{-2}yr^{-1}}$). In PEL, the snow accumulation ranged between 156–390 $\mathrm{kg\,m^{-2}yr^{-1}}$ (mean 276 $\mathrm{kg\,m^{-2}yr^{-1}}$) in the coastal section, while in the inland section it ranged between 138–376 $\mathrm{kg\,m^{-2}yr^{-1}}$ (mean 261 $\mathrm{kg\,m^{-2}yr^{-1}}$).

Stable isotope values and related parameters for snow core sites from both cDML and PEL transects are provided in Tables

1 and 2. All snow cores from both transects showed clear seasonal variations in stable isotope values except for the cores at 240 km in cDML transect and 70 and 100 km in PEL transect, where seasonality could not be established, presumably due to core damage during transportation.

The relationship between $\delta D$ and $\delta^{18}O$ from all snow cores were assessed using linear regression analysis (Fig. 2). The analysis showed a strong correlation between $\delta D$ and $\delta^{18}O$ in both cDML (orange circles) and PEL (green circles) transects

from coastal to inland region. The slope of the PEL transect (8.12) was close to that of the global meteoric water line (Craig, 1961) with the equation $\delta D = 8.12 \times \delta^{18}O + 7.10$ while the cDML transect followed the linear equation $\delta D = 7.9 \times \delta^{18}O + 2.66$. The relationship between average $\delta^{18}O$ values and annual mean temperature (RACMO 2.3 values) at each snow core site in both transects showed linearity with equations $\delta^{18}O = 0.70 \times T - 14.75$ in cDML region and $\delta^{18}O = 1.01 \times T - 5.70$ in PEL (Fig. 3).

Deuterium excess (d) was calculated for each 5 cm sub-sample of all the snow cores. The d values varied between –1.97‰ and 17.18‰ in cDML and –3.43‰ and 11.10‰ in PEL region. The relationship between near-surface temperature and 'd' was analysed using linear regression method which showed a significant relationship for samples from cDML region with the regression equation: $d = -0.11 \times T + 2.78$. (R = –0.41, p < 0.01). For the PEL region however, no significant relationship was found (Fig. 4).

Results from the isotope firn diffusion calculations using the R package by Münch and Laepple (2018) revealed a diffusion length of 6 cm over a period of 5 years (Fig. 5).

Five-day back-trajectory frequency maps of coastal, mountainous and inland locations showed vast differences in the sources between summer and winter (Fig. 6). During winter, the air parcels to cDML coast arrived from Weddell Sea and south Atlantic ocean while during summer the trajectories were mostly arriving from the Indian Ocean. Similarly, the air parcels to cDML

inland arrived from the Weddell Sea, south Atlantic and the Indian Ocean. The air mass to PEL arrived predominantly from south Indian Ocean during summer and winter.



## 4   Discussion

### 4.1   Spatial variability of snow accumulation and stable water isotopes

Spatial variations of snow accumulation in Antarctica are primarily due to the presence of physical barriers during snowfall and snow redistribution post deposition (Melvold et al., 1998; Vaughan et al., 1999). The accumulation rates in cDML region showed a large spatial variation with the near-coastal section having a substantially high average accumulation of 354 $\mathrm{kg\,m^{-2}\,yr^{-1}}$ (Table 1). Compared to this, the interior section of the transect had less than half of the accumulation rate than the coast (average 155 $\mathrm{kg\,m^{-2}\,yr^{-1}}$), while the mountainous section showed moderately high accumulation rate (average 235 $\mathrm{kg\,m^{-2}\,yr^{-1}}$). Such large spatial variability in cDML region can be attributed to the presence of extensive mountain chains existing parallel to the coast. These mountain chains in cDML act as a physical barrier to the air masses arriving from the Southern Ocean impacting the snow accumulation and redistribution. As a result, the study area could be separated into three distinct accumulation regimes. The physiography and topography of the cDML region evidently influenced the snow accumulation rates showing a strong correlation with distance and elevation (Table 3). On the contrary, the PEL transect showed moderately high accumulation with little variation between the coastal (276 $\mathrm{kg\,m^{-2}\,yr^{-1}}$) and the inland (260 $\mathrm{kg\,m^{-2}\,yr^{-1}}$) sections. Although there exist substantial slope ($> 8 \mathrm{\ m\,km^{-1}}$) in the coast of PEL (Mahalinganathan et al., 2012), it did not affect the overall accumulation rates in the region. Such uniform accumulation pattern in PEL transect could be explained by snowdrift and redistribution induced by strong katabatic winds in this region (Allison, 1998; Ding et al., 2020). With no orographic barriers to influence the flow of katabatic winds, the accumulation pattern in PEL tend to get smoothened by the wind scouring and snow redistribution.

The mean values of $\delta^{18}$O and $\delta$D from each core location decreased from coastal to inland in both cDML and PEL regions suggesting the typical continental effect, i.e. depletion of $\delta^{18}$O and $\delta$D with the increasing distance and elevation from the moisture source (Tables 1 and 2). However, the multiple regression models using the geographical parameters and $\delta^{18}$O showed negligible variance with distance and elevation (Table 4). Based on these regressions in cDML region, the snow accumulation changes were found to be the primary driver for spatial variations in $\delta^{18}$O and $\delta$D values. Surface temperature also had noticeable effect on isotopic values in this region (Table 4). However, in the PEL region, the snow accumulation did not play any role in the spatial variations of $\delta^{18}$O and $\delta$D while temperature played a crucial role. The distance from coast and elevation are known to impact the isotopic composition variation in Antarctic snow (e.g. Huybrechts et al., 2000; Masson-Delmotte et al., 2008). However, there are large regional variations and within short transects, such relationships may not be linear. Since accumulation and temperature parameters are directly impacted by the distance from the coast and elevation, they have indirect influence on the isotopic change.

The relationship between $\delta^{18}$O and $\delta$D was determined using all the snow cores and the local meteoric water lines (LMWL) for this region were calculated (Fig. 2). The $\delta^{18}$O values in cDML region (orange circles) were more depleted than that of the PEL region (green circles). Though the coastal sections in cDML and PEL appears to be almost on the same latitude ($\sim 69°$S), a difference of 6 ‰ was observed between the coasts. This could be due to the fact that the sampling in cDML begin from 110 km inland while the sampling in PEL begins 10 km from the open ocean. The slope of the LMWL in cDML (7.9) is lower



than that of the global meteoric water line (GMWL) while the slope of LMWL in PEL (8.12) had a slope close to GMWL. The cDML transect, therefore, have a slope closer to that of the Antarctic meteoric water line (AMWL), while the PEL transect have a slope closer to that of the GMWL. All the samples were utilized to observe the $\delta^{18}$O and $\delta$D relationship which did not appear to show any deviation from the AMWL.

The $\delta^{18}$O–temperature relationship vary depending on the location of the sampling sites. An Antarctic wide study by Masson-Delmotte et al. (2008) obtained a slope of 0.80‰ per degree Celsius, where the spatial slopes were calculated considering a radius of 400 km, thereby smoothing out the regional topography and moisture source areas. The annual $\delta^{18}$O–Temperature relationship is known to be weak in the coastal zones around Antarctica (Goursaud et al., 2019; Bertler et al., 2011; Thomas et al., 2013). Isaksson and Karlén (1994) observed a poor correlation between $\delta^{18}$O and Temperature below
1000 m a.s.l and a significant correlation above 1000 m a.s.l. The present study includes all such topographic variations as it covers a small region from coast and inland. In spite of such complications, the results from the present study area (Fig. 3) showed that with an increase of every $\delta^{18}$O‰ value, there was an increase in the temperature by 0.7°C in cDML and 1°C in PEL. Therefore, the proposed spatial slope (0.80‰ /°C) by Masson-Delmotte et al. (2008) seems to be reasonable.

The deuterium excess (d) values did not show any correlation with any geographical parameters in both cDML and PEL
transects (Table 3). An Antarctic-wide compilation of datasets by Masson-Delmotte et al. (2008) showed that even though both elevation and distance appear to generally control d, results from locations below 2000 m elevations did not produce any satisfactory correlation. The PEL transect in this study had only two sampling sites above 2000 m elevation, while the d above 2000 m altitude in cDML transect did not show any meaningful correlation with the geographic parameters. This could be due to involvement of other factors such as regional differences in moisture sources (Simmonds et al 2003). A significant
negative correlation was observed between surface temperature and d in cDML (Fig. 4). This negative correlation could be associated with various factors such as the changes in moisture sources, kinetic fractionation on ice crystals at supersaturation point (Jouzel and Merlivat, 1984) and increase of d along the distillation path (Masson-Delmotte et al., 2008).

## 4.2  Extent and impact of isotope diffusion

Stable water isotope records undergo rapid diffusion in snow and firn than in solid ice (Johnsen, 1977; Whillans and Grootes,
1985). As a result, the seasonal amplitude of these records is homogenized in firn – resulting in a considerable reduction in the summer and winter $\delta$ values. Since these records are the primary tool in understanding the relationship between isotope ratios and temperature, it is important to estimate the amount of isotope diffusion.

In order to understand the extent of such diffusion in a high accumulation region like the cDML transect, we compared the isotope records of a snow core (cDML 9) with that of a shallow ice core close to the snow transect, IND–33/B8 (Fig. 1). This
ice core was drilled 5 years after the snow cores were retrieved (2013–14 Summer). The detailed stable isotope records and chronology of this ice core is discussed in an upcoming paper (Tariq et al., 2020, unpublished). A direct comparison of $\delta^{18}$O values of the ice core to a snow core from the transect showed a considerable amount of diffusion in the seasonal amplitude of $\delta^{18}$O (Fig. 5). The snow cores originally had an amplitude >5‰ while the ice core seasonal amplitude was reduced to 2‰ .



Further, to analyse the extent and impact of isotope diffusion in this core site, the diffusion length was estimated using a firn
diffusion model (Johnsen et al., 2000), using the site specific temperature (–22°C), surface firn density (340 $\mathrm{kg\,m^{-3}}$), local
mean surface pressure (800 mbar) and the local accumulation rate (300 $\mathrm{kg\,m^{-2}yr^{-1}}$). The degree of smoothing of the stable
water isotope record depends on the isotopic diffusion length which is shown to increase in the topmost firn layer (Johnsen
et al., 2000). Differential diffusion ($\sigma$) was calculated using the equation $\sigma = \sqrt{\sigma^2(z_2) - \sigma^2(z_1)}$, where $z_1$ and $z_2$ are initial and
final depths (Münch et al., 2016; Münch and Laepple, 2018). Results from firn diffusion model on the 33/B8 ice core showed
a diffusion length of around 6 cm in the surface to a depth of 4 m (Fig. 5, inset). The calculation showed a rapid increase in
diffusion length from the surface before attaining a maximum at 30 meters. This findings can be used during interpretation of
stable isotope record of deep ice cores by removing the effects of diffusion on inter-annual variability.

### 4.3 Sources of moisture for precipitation in the region

The sources of moisture in Antarctica is widely distributed and strongly influenced by topography, sea-ice conditions and mid-
latitude land-ocean contrasts (Sodemann and Stohl, 2009). Moisture source regions could be identified using several techniques.
For instance, d was used as a tracer in isotope models by Petit et al. (1991) and Ciais and Jouzel (1994). An attempt was made
to understand the moisture origin by tracing backwards in time, the air parcels from different sampling sites.

Five-day air parcel backward trajectory frequency map showed moisture source regions close to 60°S latitude during summer
and up to 50°S latitude during winter (Fig. 6). The cDML region had moisture sources from the Weddell sea and south Atlantic
Ocean during winters while during summer, the sources were much restricted to the southern Atlantic and Indian Ocean. This
observation is also in line with the results from the cDML region (Rahaman et al., 2016). The winter (JJA) trajectories had
a much wider spread in comparison to summer (DJF) due to the storminess in the southern hemisphere in winter.Also, the
expanding sea ice conditions during the winter blocks the local moisture source contributions to these regions (Sodemann
and Stohl, 2009). About 90% of the trajectories converged along the coastal patch covering predominantly ice shelves, ice
rises and the ice cap itself with a small amount covering the sea-ice and open ocean region, strictly followed the orography
of this region with the southern bounds up to various mountain ranges (Queen Fabiola, Sir Rondane, Wohlthat mountains)
along 72°S. In summer, this converging zone with 90% trajectories extended up to 42°E (Prince Olav coast). As a result of
more widespread trajectory distribution in winter, this zone shifted slightly westward extending between 35°E and 8°E. About
50% of the trajectories from coastal section of cDML transect had a wider distribution covering 15°W–70°E and 60°–75°S
while a small number of trajectories (∼10%) originated beyond 55°S from the Southern Ocean and 75°S in the continent. In
winter, the trajectories had a much wider coverage extending between 30°W–70°E and 57°–80°S. Similarly, trajectories from
the coastal and inland regions of PEL showed a predominant distribution towards east of the transect with 90% of trajectories
converging up to 100°E (Shackleton ice shelf) in summer and up to 110°E in winter. The southern bounds of the trajectories
were controlled by steep orography – with 2000 m elevation limit. In summer, about 50% of the trajectories in coastal section of
PEL extended between 60°–75°S and 50°–140°E with an increase of about 5° spread in winter. The inland section of transect
had moisture sources all the way between 60°–80°S. In summary, the moisture sources in both cDML and PEL regions were
strongly controlled by orography, with highly mixed sources originating from the Southern Ocean.



# 5 Conclusions

Our study explored the spatial variations in snow accumulation, stable isotopic variations, the moisture sources as well as the

220 post-depositional isotope diffusion in coastal Antarctica. The snow accumulation in central Dronning Maud Land (cDML) region was affected by the mountainous topography while in the Princess Elizabeth Land (PEL) region accumulation pattern smoothed due to the absence of orographic influence. The stable water isotopes from high resolution snow cores sampled from the same season from different regions in Antarctica aided in understanding the spatial variations and to investigate the influences of meteorological and geographical controls. The $\delta^{18}$O–T relationship showed a relationship of 0.7‰ °C$^{-1}$

in cDML and 1‰ °C$^{-1}$ in PEL. Considering the spatial variations in $\delta^{18}$O–T gradients and the complex difference between snow and firn samples within the continent, careful consideration should be exercised during the application of paleoclimate reconstruction. Comparison of snow core isotope data with the firn core revealed a considerable diffusion with time, showing a diffusion length of about 6 cm from the surface to the depth of 4 m in 5 years. Moisture source studies in this region showed that the majority of the air masses to cDML originated from the south Atlantic (up to 50°S) and Weddell sea during winter and

mainly south Atlantic in summer. Within the PEL region, the source was mainly from the Indian Ocean sector.

*Author contributions.* All authors contributed equally to the work presented in this paper. K. Mahalinganathan collected snow cores and the ice core, analysed data and wrote the paper. Tariq Ejaz and Redkar B.L. analysed the samples. Laluraj C.M. analysed data and contributed to the development of this manuscript. M. Thamban collected the ice core, analysed data and wrote the paper.

*Competing interests.* The authors declare that the research was conducted in the absence of any commercial or financial relationships that

could be construed as a potential conflict of interest.

*Acknowledgements.* We thank the director of the National Centre for Polar and Ocean Research for his encouragement. Ministry of Earth Sciences is thanked for financial support under the project "Cryosphere and Climate". We are grateful for the support from the members and logistic crew of the 28th and 33rd Indian Scientific Expedition to Antarctica. We thank Dr. Thomas Münch for the R packages to estimate firn diffusion length. We also thank the Norwegian Polar Institute for the Quantarctica QGIS package. This is NCAOR contribution number

xx/xxxx.



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





# Figures

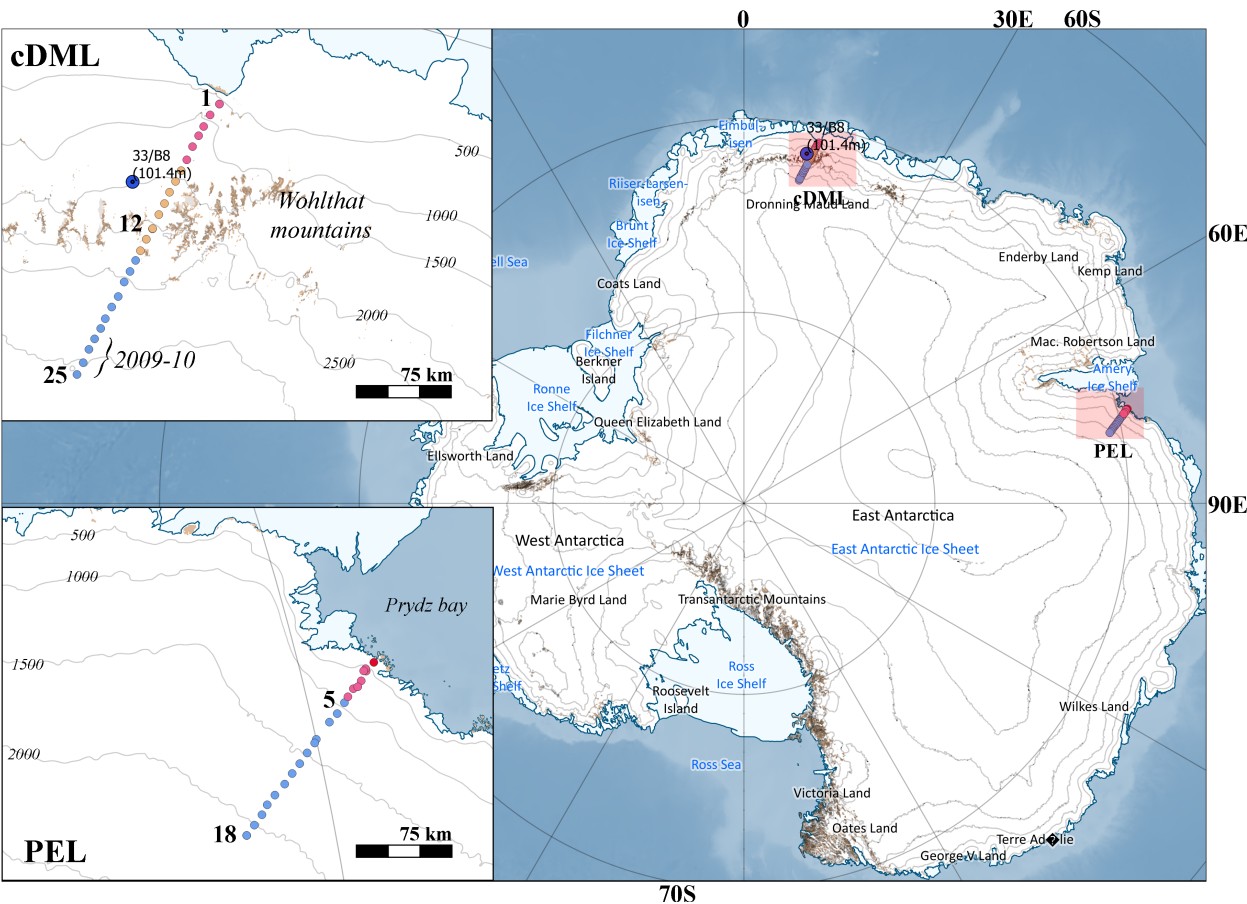

**Figure 1.** Study region showing sampling locations along the central Dronning Maud Land (cDML) and Princess Elizabeth Land (PEL) in East Antarctica. Colour coded sampling locations indicate coastal (red), mountainous (orange) and inland (blue) regions of the transect. The ice core 33/B8 shown in the cDML region was drilled during 2013–14 field season. Background map sourced from Hillshades and Elevation model compiled using ETOPO1, IBSCO and RAMP2 data.
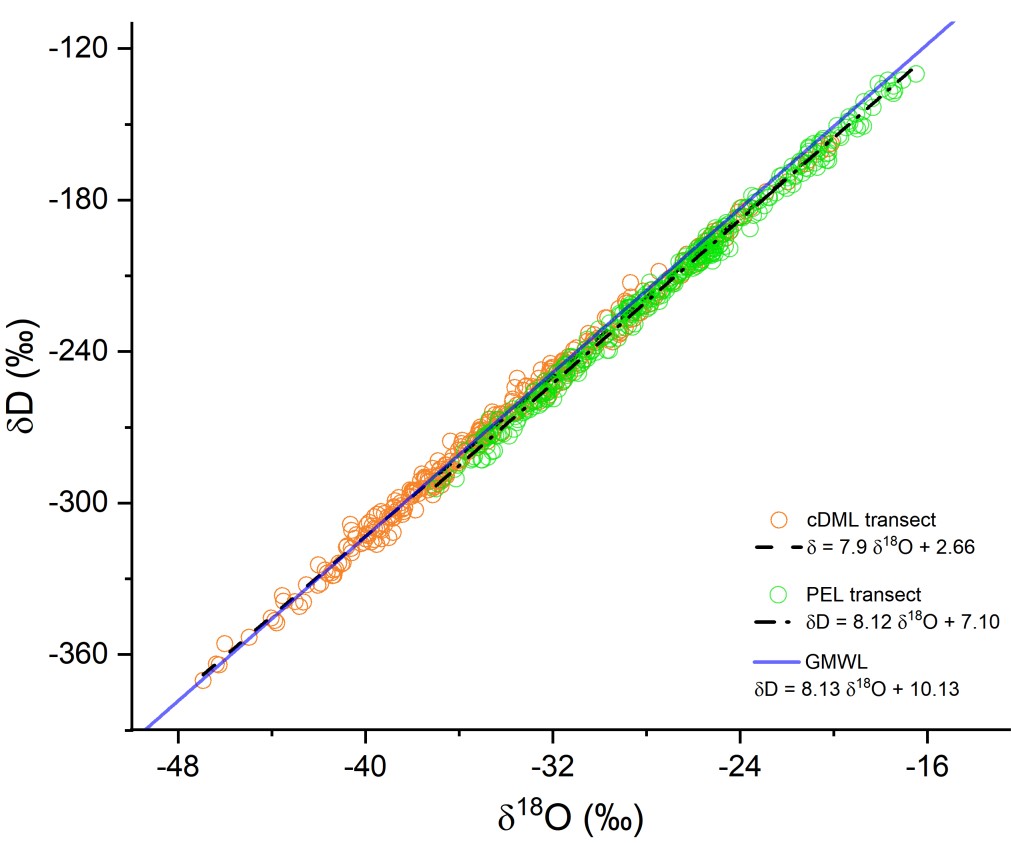

**Figure 2.** Relationship between $\delta D$ and $\delta^{18}O$ in cDML and PEL transects. Blue line is Global Meteoric Water Line.

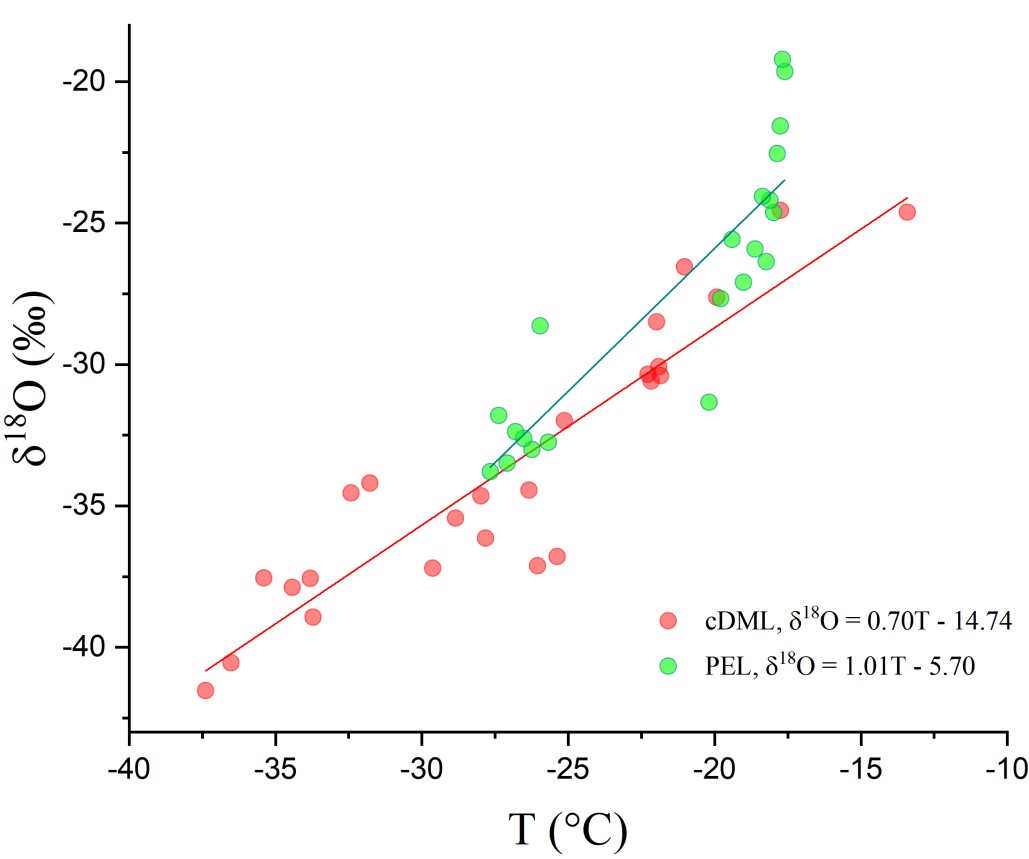

**Figure 3.** Relationship between mean annual $\delta^{18}O$ values and annual mean temperature in cDML and PEL transects.

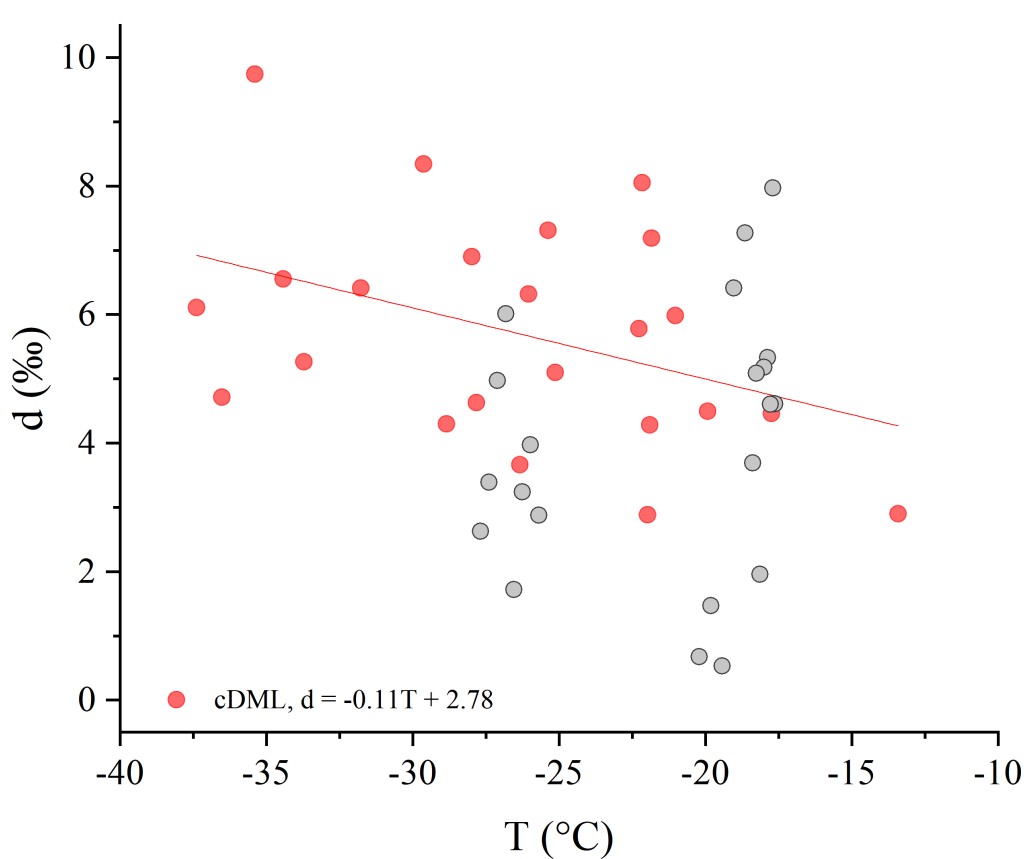

**Figure 4.** Relationship between deuterium excess and Temperature from cDML and PEL regions.



**Figure 5.** Differences in seasonal amplitude of $\delta^{18}O$ in snow core and ice core. The ice core was drilled after 5 seasons in 2013–14 summer. The firn diffusion model (inset) calculated based on site specific temperature, pressure and density of surface firn showed a diffusion length of around 6 cm in the surface explaining the amplitude diffusion in ice core.





**Figure 6.** Five-day back-trajectory frequencies in coastal (top), inland (middle) and mountainous sections (bottom panels) of the cDML and PEL transects during the austral summer (left panel) and winter (right). The translucent purple color shows the lowest frequency and the translucent green shows the highest frequency of trajectories arriving to the sampling sites. The frequencies were derived using HYS-PLIT model with inputs from meteorological parameters of GDAS dataset. Background map sourced from Hillshades and Elevation model compiled using ETOPO1, IBSCO and RAMP2 data.





345 **Tables**

**Table 1.** Geographical distribution and averaged stable isotope values of the snow cores from cDML region.

| cDML | Lat S | Lon E | Dis km | Ele m | $\delta D$ ‰ | $\delta^{18}O$ ‰ | d ‰ | Temp °C | Acc $kg\,m^{-2}\,yr^{-1}$ |
|---|---|---|---|---|---|---|---|---|---|
| 1 | −70.85 | 11.74 | 110 | 610 | −193.97 | −24.61 | 2.90 | −13.4 | - |
| 2 | −70.94 | 11.59 | 120 | 914 | −191.90 | −24.55 | 4.46 | −17.8 | 361 |
| 3 | −71.03 | 11.53 | 130 | 1067 | −216.44 | −27.62 | 4.5 | −19.9 | 352 |
| 4 | −71.12 | 11.45 | 140 | 1158 | −206.37 | −26.54 | 5.99 | −21 | 313 |
| 5 | −71.21 | 11.37 | 150 | 1250 | −235.92 | −30.39 | 7.19 | −21.8 | 334 |
| 6 | −71.3 | 11.3 | 160 | 1341 | −236.31 | −30.07 | 4.28 | −21.9 | 410 |
| 7 | −71.38 | 11.21 | 170 | 1524 | −236.57 | −30.58 | 8.05 | −22.2 | 320 |
| 8 | −71.46 | 11.14 | 180 | 1600 | −225 | −28.49 | 2.88 | −22 | 227 |
| 9 | −71.54 | 11.06 | 190 | 1768 | −236.95 | −30.34 | 5.78 | −22.3 | 157 |
| 10 | −71.64 | 10.99 | 200 | 1890 | −250.79 | −31.99 | 5.1 | −25.1 | 226 |
| 11 | −71.73 | 10.91 | 210 | 2012 | −271.83 | −34.44 | 3.66 | −26.3 | 224 |
| 12 | −71.81 | 10.83 | 220 | 2103 | −287.01 | −36.79 | 7.31 | −25.4 | 202 |
| 13 | −71.89 | 10.75 | 230 | 2225 | −290.59 | −37.11 | 6.32 | −26 | 292 |
| 14 | −71.98 | 10.67 | 240 | 2377 | −270.23 | −34.64 | 6.9 | −28 | - |
| 15 | −72.07 | 10.6 | 250 | 2576 | −284.43 | −36.13 | 4.63 | −27.8 | 106 |
| 16 | −72.16 | 10.51 | 260 | 2652 | −279.1 | −35.43 | 4.3 | −28.8 | 180 |
| 17 | −72.23 | 10.43 | 270 | 2743 | −289.24 | −37.2 | 8.35 | −29.6 | 207 |
| 18 | −72.33 | 10.34 | 280 | 2865 | −267.1 | −34.19 | 6.41 | −31.8 | 222 |
| 19 | −72.41 | 10.25 | 290 | 2896 | - | - | - | −32.4 | 249 |
| 20 | −72.5 | 10.16 | 300 | 3018 | - | - | - | −33.8 | 157 |
| 21 | −72.57 | 10.07 | 310 | 3045 | −306.2 | −38.93 | 5.27 | −33.7 | 126 |
| 22 | −72.65 | 9.97 | 320 | 3068 | −296.47 | −37.88 | 6.55 | −34.4 | 123 |
| 23 | −72.73 | 9.88 | 330 | 3082 | −290.59 | −37.54 | 9.74 | −35.4 | 117 |
| 24 | −72.83 | 9.8 | 340 | 3101 | −319.65 | −40.55 | 4.71 | −36.5 | 110 |
| 25 | −72.92 | 9.7 | 350 | 3130 | −326.09 | −41.53 | 6.11 | −37.4 | 104 |





**Table 2.** Geographical distribution and averaged stable isotope values of the snow cores from PEL region.

| PEL | Lat S | Lon E | Dis km | Ele m | $\delta$D ‰ | $\delta^{18}$O ‰ | d ‰ | Temp °C | Acc $\mathrm{kg\,m^{-2}\,yr^{-1}}$ |
|---|---|---|---|---|---|---|---|---|---|
| 2 | −69.55 | 76.3 | 20 | 300 | −152.55 | −19.64 | 4.61 | −14.2 | 227 |
| 19 | −69.55 | 76.25 | 23 | 579 | −145.71 | −19.21 | 7.97 | −14.4 | 156 |
| 20 | −69.56 | 76.28 | 26 | 640 | −167.94 | −21.57 | 4.6 | −14.6 | 269 |
| 3 | −69.6 | 76.46 | 30 | 792 | −175.02 | −22.54 | 5.33 | −16.1 | 389 |
| 21 | −69.63 | 76.56 | 35 | 884 | −191.87 | −24.63 | 5.18 | −16.9 | 200 |
| 4 | −69.67 | 76.58 | 40 | 975 | −191.58 | −24.19 | 1.95 | −17.2 | 329 |
| 22 | −69.71 | 76.71 | 45 | 1036 | −205.81 | −26.36 | 5.08 | −17.8 | 292 |
| 5 | −69.75 | 76.81 | 50 | 1113 | −188.78 | −24.06 | 3.69 | −18.5 | 347 |
| 6 | −69.81 | 77 | 60 | 1234 | −200.06 | −25.92 | 7.27 | −19.3 | 270 |
| 7 | −69.88 | 77.13 | 70 | 1280 | −210.32 | −27.09 | 6.41 | −19.8 | |
| 8 | −70 | 77.41 | 87 | 1494 | −204.08 | −25.58 | 0.53 | −21.1 | 376 |
| 9 | −70.01 | 77.48 | 90 | 1509 | −219.93 | −27.67 | 1.47 | −21.3 | 335 |
| 10 | −70.08 | 77.65 | 100 | 1585 | −250.04 | −31.34 | 0.67 | −21.8 | |
| 11 | −70.15 | 77.83 | 110 | 1631 | −259.17 | −32.76 | 2.87 | −22.3 | 256 |
| 12 | −70.21 | 78 | 120 | 1722 | −225.12 | −28.64 | 3.97 | −22.8 | 138 |
| 13 | −70.28 | 78.18 | 130 | 1768 | −260.84 | −33.01 | 3.23 | −23.2 | 249 |
| 14 | −70.36 | 78.36 | 140 | 1875 | −259.26 | −32.62 | 1.71 | −24 | 179 |
| 15 | −70.43 | 78.53 | 150 | 1920 | −253.01 | −32.38 | 6.01 | −24.5 | 335 |
| 16 | −70.48 | 78.71 | 160 | 1987 | −262.96 | −33.49 | 4.97 | −25 | 225 |
| 17 | −70.55 | 78.9 | 170 | 2118 | −251.04 | −31.8 | 3.39 | −25.8 | 246 |
| 18 | −70.61 | 79.08 | 180 | 2210 | −267.71 | −33.79 | 2.62 | −26.6 | 263 |





**Table 3.** Correlation between stable water isotopes and physical and climate parameters in cDML and PEL regions. All correlations are significant at 0.01 level.

|  | Distance | Elevation | $\delta$D | $\delta^{18}$O | d | Temp | Acc |
|---|---|---|---|---|---|---|---|
| | | | central Dronning Maud Land | | | | |
| Distance | 1 | 0.986 | –0.936 | –0.937 | - | –0.985 | –0.834 |
| Elevation | 0.973 | 1 | –0.937 | –0.939 | - | –0.974 | –0.834 |
| $\delta$D | –0.925 | –0.946 | 1 | 0.999 | - | 0.923 | 0.781 |
| $\delta^{18}$O | –0.927 | –0.946 | 0.999 | 1 | - | 0.926 | 0.778 |
| d | - | - | - | - | 1 | - | - |
| Temp | –0.983 | –0.995 | 0.948 | 0.947 | - | 1 | 0.794 |
| Acc. | - | - | - | - | - | - | 1 |
| | | | Princess Elizabeth Land | | | | |

**Table 4.** Multiple regression model summaries explaining the percentage of stable isotope variability attributed to accumulation, temperature and the physical parameters. Temperature effect is clear in PEL while precipitation effect is dominant in cDML region.

| PEL Model | | $\delta^{18}$O | |
|---|---|---|---|
| | | R | Variance (%) |
| 1 | Accumulation | 0.109 | 1.2 |
| 2 | Accumulation, Temperature | 0.952 | 89.4 |
| 3 | Acc., Temp., Distance | 0.952 | 0.1 |
| 4 | Acc., Temp., Distance, Elevation | 0.953 | 0 |
| cDML Model | | | |
| 1 | Accumulation | 0.778 | 60.5 |
| 2 | Accumulation, Temperature | 0.915 | 23.2 |
| 3 | Accumulation, Temperature, Distance | 0.930 | 2.9 |
| 4 | Acc., Temp., Distance, Elevation | 0.934 | 0.7 |