# Peer review of "Spatial distribution and post-depositional diffusion of stable water isotopes in East Antarctica"

_The Cryosphere, 2020_

## Referee Comment (RC1) · Anonymous Referee #1 · 6 May 2020

This manuscript presents results about superficial snow cores across the Dronning Maud Land and Princess Elizabeth Land, as well as a superficial ice cores from near the DML transect. The produced data are of high value as they can help identify in two coastal areas of East Antarctica if the deposition leading to the formation of the firn is homogenous over areas characterised with largely changing conditions (temperature, accumulation…). This type of study is very valuable because they will help in the long run to improve the interpretation of the water isotopic composition from ice core records.

At this stage though, I cannot recommend that the manuscript is accepted for publica-

tion. While I'm really keen on seeing these results published, I believe that the current version of the manuscript present significant flaws which do not honour the high quality of the dataset which was I'm sure produced at high costs (at least 3 Antarctic campaigns).

I have included some more specific points as major and minor comments on the manuscript. In general, I do not believe that the manuscript is providing any critical interpretation of the results. While the authors compared their results to various other results from past publications (for instance the diffusion models of (Münch, Kipfstuhl, Freitag, Meyer, Laepple, 2017) or the isotope-isotopes or isotope-temperature slopes from (Masson-Delmotte et al., 2008)), they fail to include the added values obtained by the comparison, or to bring up new interpretation as a scientific paper should do. In the current state, the manuscript provides and makes public this very valuable dataset, but, in my opinion, is not using it to learn something that could help the scientific community. It saddens me to give this harsh review to this manuscript which I would have liked to see through. I sincerely hope that my comments below will help the authors in the complex interpretation of this very valuable dataset.

Major Comments:

The introduction could be improved. It is difficult to follow what is the goal of the introduction in respect with the abstract, and the title of the article. At this stage, I could expect anything from a review on water isotopes in Antarctica to the interpretation of a new ice core record. Overall the manuscript seems to cover a wide range of different aspect of what could be done with these data, but do not actually follow through on any of these. I would suggest to refocus the introduction, and the manuscript on the main point of the manuscript, which in my opinion is the comparison between the two transects and the comparison between the ice core and the cDML transect.

One major aspect comes from the evaluation of the accumulation rates for the difference sites. If I understood well the paper, they were estimated from the successive

maxima/minima from the isotopic profiles, yet, there is evidence that diffusion of white noise would also create successive cycles that look similar to the seasonal cycle (Laepple et al., 2018). I think it would be very valuable for the manuscript to have rigorous tests that the accumulation rates determined here are not artefacts created by the diffusion length. Typically, a good test would be to diffuse with noise for each site and compare the mean distance between maxima/minima of the diffuse core with the one of the observed core. If the accumulation rate predicted for the diffused white noise is close to the observed accumulation, this would cast doubts on the determination of the accumulation rate in my opinion. Typically, we have been able to observe this phenomenon for accumulation rates up to 80 kg.m-2.a-1 (w.e.), and have never tried this for accumulation as high as observed here. Yet, this could also be affecting the results here for sites such as cDML 15, 24, 25. Observing the results from Fig. 5 where the diffusion length is roughly 6cm for the first meters (lines 110 to 111), one would expect white noise diffused cycles of 5 times the diffusion length, so roughly 30cm, which seems rather similar to what is observed around 4m deep in Fig. 5. Also, are the accumulation rates in water equivalent ? As this will have a strong impact on the diffusion.

The interpretation of the multiple regression model seems to be arbitrary to me in the present form. In Table 3 one can see the correlation between the different variables (even though the part of the table for PEL seems to be missing which prevents from interpreting Table 4). In cDML, roughly all the variables are very well correlated with one another, as a result, when you do the multiple regression model in Table 4, you're not proving that the distance and the elevation explain negligible variance, but that the variance explained by the temperature and the accumulation is most likely linked already to the distance and the elevation. The argument for the entire paragraph from line 135 to 145 seems completely specious to me.

The comparison of the ice core and the snow cores is probably the most important aspect that could be included in this manuscript. At this stage, it is not possible to

evaluate how much the signal got dampened considering how different appear to be the sites of the cDML9 and the ice core drilling site. The accumulation of the ice core drilling site appears to be twice as large as of cDML9, while this is not mentioned anywhere in the manuscript. Overall, "I have the feeling" that the ice core drilling site would match what the authors refer to as a "coastal" type of site, while it's compared to cDML9 which is classified as "mountainous". First, I should not have to guess or "have a feeling", and this should be studied in the manuscript. In general, considering that you have 25 snow cores, I feel it would make sense to align all of them to the core and to have statistical constrains on the original amplitude, as the sites were actually different (25km apart I would guess from Fig. 1). Second, there is no clear constrain of post-depositional processes here, not any use of the comparison between the snow core and the ice core. For instance, can you use this method to actually deconvoluate the diffusion from the climatic signal and reconstruct the temperature from the firn core ? If so, does it compare with the model temperature time series for this site ? Only if you are actually able to do this have you properly constrained the post depositional processes. Also, for a coastal site like the one of the ice core, I have trouble to imagine that diffusion is the most important post-depositional process affecting your ice core, compared to sublimation/condensation combined with katabatic winds, wind redistribution or scouring. . .

Minor Comments:

Lines 58 to 60: "After recovery, the snow cores were transferred directly into pre-cleaned high-density polyethylene bags and sealed immediately to avoid contamination during storage and shipped under frozen conditions to the National Centre for Polar and Ocean Research, India. The cores were stored at –20°C till analysis." Is there any evidence that diffusion cannot happen in highly porous snow at -20°C ? These conditions are very close to summer firn conditions at places like Dome C, where diffusion still takes place.

Lines 75 to 77: "The seasonality in snow cores was determined by establishing the

summer and winter peaks in isotope records where a minimum amplitude of 4‰ between summer and winter was used to differentiate these peaks as detailed in figure 2 our previous publications (Mahalinganathan et al., 2012; Mahalinganathan and Thamban, 2016)." There are evidences that the cycles observed in isotopic profiles of water isotopes can be linked to diffused white noise, especially for low accumulation areas (Laepple et al., 2018), as a result, they can be deceived when used to identify annual layers and date ice cores. It is not necessarily going to be the case for snow cores that close to the coast, but it's definitely something good to discuss. Lines 88 to 93: Are the accumulation rates obtained from the distance between maxima/minima in the isotopic profiles ? In which case, as mentioned in my previous comment, I'd strongly suggest to provide evidence that similar distance would not be created artificially by diffusion of white noise.

Lines 112 to 113: "Five-day back-trajectory frequency maps of coastal, mountainous and inland locations showed vast differences in the sources between summer and winter (Fig. 6)" Are they vastly different? I'm a bit confused for several reasons. First, what months did you choose for summer and winter ? Antarctic summer is rather brief for a lot of sites, with a large asymmetry between summer months (DJF), and winter months (AMJJASO) (Van Den Broeke, 1998). As a result, statistically, you could already explain having less events in summer, and as a result, a small range of possible storms visible in the sample. Second, it seems that if the dark red and dark green envelopes do cover a large area, most of the trajectory originates from very similar cones both in summer and in winter, which is quite unexpected, and very intriguing. Considering that the dark red corresponds most likely to a very small number of trajectory, it would be interesting to have quantitative numbers of trajectories for each of the sectors you mention here, to actually be able to evaluate information beyond the plot. Finally, this does not necessarily reflect the actual contribution to the different ice cores, indeed, a small number of events can contribute a large amount of the accumulation, and I have serious doubts that the trajectories originating from the Plateau contribute for a lot of the accumulation for instance. I would recommend to realise an analysis similar to the

one found in Figure 5 of (Genthon, Six, Scarchilli, Ciardini, Frezzotti, 2015) and include the relative contributions to the snow fall amount for clusters of trajectories.

Lines 119 to 120: "Spatial variations of snow accumulation in Antarctica are primarily due to the presence of physical barriers during snowfall and snow redistribution post deposition (Melvold et al., 1998; Vaughan et al., 1999)." I would say I disagree with this sentence, and that there are a lot of literature that has been produced since 1999 going in a completely different direction. Depending on your definition of "spatial variations of snow accumulation", you can obviously interpret this sentence in a lot of different ways, so if you look at small scales, it's been shown that at Dome C, the accumulation can vary by large amount over short distance without any physical barriers like montains (Genthon et al., 2015). Looking at typical coast-to-interior patterns, I would recommend for instance to consider (Agosta et al., 2019) which shows that even for other sites without mountain ranges, the dominant pattern is the coast-to-plateau gradient.

Lines 125 to 130: "These mountain chains in cDML act as a physical barrier to the air masses arriving from the Southern Ocean impacting the snow accumulation and redistribution. As a result, the study area could be separated into three distinct accumulation regimes. The physiography and topography of the cDML region evidently influenced the snow accumulation rates showing a strong correlation with distance and elevation (Table 3). On the contrary, the PEL transect showed moderately high accumulation with little variation between the coastal (276 kgm$-2$ yr$-1$) and the inland (260 kgm$-2$ yr$-1$) sections." While it's true that it seems that the accumulation gradient is larger for the DML site, and that one could expect the mountain range to affect the accumulation along the transect, I don't think you've proven it yet. Indeed, the PEL transect does not reach altitude as high as the cDML transect, which could also explain the difference in accumulation. As previously mentioned, I would also be careful with the evaluation of the accumulation rates from the isotopes, considering you're using the temperature from RACMO, I would suggest to also include the accumulation rates from RACMO

which might help in the interpretation here (similar to (Agosta et al., 2019)), or to even remove this paragraph.

Lines 137 to 138: "However, the multiple regression models using the geographical parameters and 18O showed negligible variance with distance and elevation (Table 4)." Considering how well correlated are the temperature, the elevation and the distance to the coast, I don't think you can make this assumption here.

Lines 150 to 151: "The slope of the LMWL in cDML (7.9) is lower than that of the global meteoric water line (GMWL) while the slope of LMWL in PEL (8.12) had a slope close to GMWL." It would help here to have error bars on the MWL slope, as well as correlation coefficient and significance tests to assess the robustness of the variations of the slopes of 0.1 around the MWL.

Lines 163: "Therefore, the proposed spatial slope (0.80‰ /C) by Masson-Delmotte et al. (2008) seems to be reasonable." I would say that the slopes you obtain seem reasonable compared to the ones found in (Masson-Delmotte et al., 2008), considering that in the aforementioned article were included over 1000 snow pits and ice cores, across all over Antarctica. The two sentence are equivalent as the statement is a bijection, but it seems more that your results are validated by what was already found in this study, than you are validating this previous study considering the content provided in both cases.

Lines 180 to 181: "The detailed stable isotope records and chronology of this ice core is discussed in an upcoming paper (Tariq et al., 2020, unpublished)." Considering that the synchronisation of both cores is key, while the Tariq et al is not available to evaluate how the two records was synchronised, it is difficult to evaluate the work in the section. Indeed, the accumulation for cDML9 is 157 kg.m-2.a-1, assuming that the value is in water equivalent (which it ought to be), this means that in 5 years, you expect 2.75m of accumulation, very far from what is shown in Fig. 5. As the core is still quite far from the site, it is possible that the accumulation was slightly different, and the value

[Figure]

of cDML9 seems to be much lower than the neighbouring sites, so even taking into account the values for cDML8 and 10, we would expect 3.94m, quite short to what is described in Fig 5. If the accumulation rates at the ice core site are that different from the accumulation rates along the transect, can you provide evidence that the amplitude of seasonal cycle of isotopic composition in the precipitation was the same? Typically, considering that the cDML transect sites neighbouring the ice cores are all mountainous sites, while the ice core seems to be in a more "coastal" sites, can you illustrate if you obtain similar amplitudes in the different firn cores of the transect that would justify that the original amplitude of the ice core could reasonably be close to what was in cDML9. Considering the large difference of amplitude between what is found for cDML9 (>5permil) and the surface of the ice core in summer/winter 2012-2013 (<2permil), one could also wonder if the ice core site just has singularly less pronounced seasonal cycle of precipitation isotopic composition.

Figure 5: Which snow core is included in the figure ? I couldn't find the information easily.

Bibliography

Agosta, C., Amory, C., Kittel, C., Orsi, A., Favier, V., Gallée, H., . . . van de Berg, W. J. (2019). Estimation of the Antarctic surface mass balance using the regional climate model MAR (1979-2015) and identification of dominant processes. Cryosphere, 13(1), 281–296.

Genthon, C., Six, D., Scarchilli, C., Ciardini, V., Frezzotti, M. (2015). Meteorological and snow accumulation gradients across Dome C, East Antarctic plateau. International Journal of Climatology, n/a–n/a. https://doi.org/10.1002/joc.4362

Laepple, T., Münch, T., Casado, M., Hoerhold, M., Landais, A., Kipfstuhl, S. (2018). On the similarity and apparent cycles of isotopic variations in East Antarctic snow pits. The Cryosphere, 12(1), 169–187. https://doi.org/10.5194/tc-12-169-2018

Masson-Delmotte, V., Hou, S., Ekaykin, A., Jouzel, J., Aristarain, A., Bernardo, R. T., . . . White, J. W. C. (2008). A Review of Antarctic Surface Snow Isotopic Composition: Observations, Atmospheric Circulation, and Isotopic Modeling*. Journal of Climate, 21(13), 3359–3387. https://doi.org/10.1175/2007JCLI2139.1

Münch, T., Kipfstuhl, S., Freitag, J., Meyer, H., Laepple, T. (2017). Constraints on post-depositional isotope modifications in East Antarctic firn from analysing temporal changes of isotope profiles. The Cryosphere, 11, 2175–2188.

Van Den Broeke, M. R. (1998). The semi-annual oscillation and Antarctic climate. Part 1: Influence on near surface temperatures (1957–79). Antarctic Science, 10(02), 175–183.
* * *

---

## Referee Comment (RC2) · Anonymous Referee #2 · 24 May 2020

Review of manuscript tc-2020-77:
"Spatial distribution and post-depositional diffusion
of stable water isotopes in East Antarctica"
by Mahalinganathan Kanthanathan et al.

24th May 2020

**Summary**

The authors present new stable isotope and accumulation data retrieved from two spatial (coast to inland) transects in the East Antarctic regions of central Dronning Maud Land (cDML) and Princess Elizabeth Land (PEL). The data were obtained from drilling and analysing short (1 m) snow profiles at 25 and 21 positions along the respective transects, covering coastal, mountaineous (cDML) and plateau regions. Based on the data, the authors present standard analyses of the relationships between oxygen and hydrogen isotopic data as well as between isotopic data and local temperature and accumulation rate, and how these vary between the two transects. In addition, they present brief diffusion and back-trajectory analyses. While the paper presents new data, the overall quality and scientific originality of this work, as I will outline in my general comments below, does not meet the standards of The Cryosphere. Therefore, I rate this manuscript as being not accetable for final publication in this journal.

**General comments**

The major shortcomings of this work are its low quality and scientific originality.

Regarding quality, the writing suffers from many small grammatical errors, which make the manuscript hard to read. Additionally, the introduction is poorly structured, the methods incomplete, and the results read like a dry technical document using many repetitive phrases. Overall, I would strongly suggest the authors to consult a language editing service to improve grammar and style. In addition, there seem to be some inconsistencies between the data as given in the tables and presented in the figures (see my specific comments).

Furthermore, the paper lacks significant scientific originality. While the authors present an extensive data set from two expensive transect sampling campaigns, which in principle would offer the chance for an interesting study, they unfortunately fail to exploit this scientific potential. What the authors present is a set of analyses which are standard for isotope studies and which already have been conducted and shown many times before. This is especially sad since the authors seem to be well aware of the challenges and uncertainties related to the interpretation of (surface snow) isotopic data in Antarctica as it has been developed by the recent literature. However, by contrast, this study does not offer any new insights which could aid the community in gaining a deeper understanding of the involved processes shaping firn and ice isotopic records. The two following examples are symptomatic for this deficiency. The authors make use of state-of-the-art firn diffusion modelling to estimate the amount of diffusion which should have attenuated an isotopic profile over the course of time, but they miss to actually test this diffusion model for their chosen site by quantitatively comparing how the model prediction fits to the data. The second example is the discussion of the back-trajectory analysis. The most text of the respective discussion section actually presents additional analysis results that were not mentioned in the results section before, while the small portion of interpretation remains

at a poorly speculative level, which leaves the reader guessing what we can learn from this analysis. An interesting aspect of the presented data set is the different importance of the spatial accumulation variability for the spatial isotope variability of the two transects. This would be an interesting candidate for an in-depth study, which the authors however do not pursue. Also, the transect data seems to comprise at least 1 m long isotope profiles for more than 40 positions, but the authors do not analyse or show any of these profiles, instead sticking to the analysis of only the mean values. Overall, the manuscript fails to meet the scientific criteria of The Cryosphere and is not in a state to be accepted for final publication.

**Specific comments**

Below I list some specific comments that could help to improve the manuscript, but I do not include the many occurrences of grammatical errors, missing specific articles, etc.

- L1: The abstract should briefly introduce the background of the study in a first sentence.

- L1: Please consider using the correct terminology "stable water isotopologues", or use phrases such as "stable isotopes" or "isotopic composition".

- L3: Please introduce $\delta^{18}$O and $\delta^2$H shortly here, or alternatively, rewrite this passage and introduce the notation in the introduction.

- L8: The term HYSPLIT is not explained anywhere here. Please also note in this regard that you do not explain how the back-trajectory analysis is actually conducted, whether in the Methods nor the results text.

- L18: Why is high resolution important in this context?

- L25: I do not see any obvious direct connection between the Town et al. study on atmosphere–surface snow exchange and wind-driven erosion and redistribution processes. You have to elaborate more on this, and consider adding more relevant recent literature, e.g. the stratigraphic noise studies by Münch et al. (2016, 2017) and the mega dune studies by Ekaykin et al. (e.g. Ekaykin et al., 2016).

- L27; sentence "The deuterium excess [...]": There is no logical link to the previous text.

- L36; "no net change [...] were observed": This statement is misleading. For normal firn diffusion and densification, we do not expect any net mean change, as a basic property of these processes. As it is written, one could interpret this as something which just has not been observed yet due to measurement uncertainty or such.

- L36: The statement that diffusion in ice is negligible has to be put into context. Ice diffusion is certainly not negligible for very deep ice cores = very old ice!

- L36; "and therefore gets preserved": Unclear from the context, what gets preserved?

- L27; sentence "Studies by...": There is no logical link to the previous text.

- L43 first sentence: This statement is in direct contradiction to the sentence about the studies by Steen-Larsen et al. (2014) and Ritter et al. (2016) a few sentences above.

- LL15-50: Overall, the introduction is poorly structured with missing logical links between sentences and paragraphs. You may consider rewriting it altogether.

- L67: What do you mean by external precision? To which isotope species do you refer here – $\delta^{18}$O or $\delta^2$H? Why do you measure samples also on the Los Gatos device, if all samples have been already analyzed on a mass spectrometer (cf. L65 " All samples...")?

– LL75-76: As you also have impurity data available from the same cores (as evident from previous publications), it would be probably more robust to conduct the dating using isotope and impurity species together, instead of setting an arbitrary threshold on the isotope peak heights.

– LL51-86: Descriptions of the HYSPLIT back-trajectory analysis and the used multiple regression model is missing.

– L95: It would be actually nice to see some example isotope profiles or even use them to further study the spatial variability and its drivers along the two transects.

– LL107-109: Using the data provided in Tables 1 and 2, I find for both transects a significant correlation between d-excess and temperature on the 0.1 significance level ($R = -0.4$, $p = 0.06$ for cDML; $R = 0.39$, $p = 0.08$ for PEL) – can you comment on this? In any case, concerning the opposite correlations and the significant scatter of the data (Fig. 4), I however ask myself what we can actually learn from regressing 'd' against temperature here and if there is actually any meaningful explanation behind the correlations?

– L137: What kind of multiple regression model do you use here? This could be added to the Methods section.

– L153: What do you mean here with "all the samples"? Did you use the combination of cDML and PEL samples for another $\delta^{18}O - \delta^2H$ regression?

– L163; "seems to be reasonable": This is expressed the wrong way around and anyway a quite obvious result: Your slope results from two Antarctic subset regions scatter around the Antarctic-wide mean slope – as it is to be expected.

– L169-172: It should be mentioned that the relationship by contrast is positive for the PEL transect data at the same significance level (according to my estimate, see the previous comment above), so a more detailed discussion of the d-excess-to-temperature relationship is needed here.

– L188-189: What exactly are the initial and final depths here?

– L189: Münch et al. (2016) is not the correct reference; I guess you mean Münch et al. (2017), where a similar approach was taken.

– Section 4.2: What do we learn from this? Have you tried to forward-diffuse the younger record with the estimated diffusion length to see if the mismatch in seasonal amplitude is really only a result of diffusion? By which value did you shift the newer data downwards and how did you choose this value? If you really want to constrain the post-depositional changes at this site similar to the study by Münch et al. (2017), you also have to take into account the effects of densification and stratigraphic noise, but I also think that the extent of overlap your two records have is much too small to arrive at any meaningful conclusions.

– Section 4.3: This section effectively presents additional results from the back-trajectory analysis, which should be placed therefore into the respective results section. Beyond that, it is pretty much unclear to me what we learn from this exercise.

– Figure 1: Please explain the abbreviations ETOPO1, IBSCO and RAMP2, and provde a source for these map data.

– Figure 3: There are two more cDML points on this figure than given in Table 1 (there are two missing isotope values in the table). For PEL, I cannot reproduce this figure using the values given in Table 2: while the overall plot looks similar, there are many offsets between the different points and also a linear regression gives slightly different results. Please carefully check that the given data are consistent with the figures and that they always reflect the latest version of your work.

– Figure 4: While I can reproduce the cDML plot given the data in Table 1, the PEL plot using the data of Table 2 looks different than the one provided here. Please check the consistency of all your data and figures; see also my comment on Fig. 3.

– Table 1; caption: What kind of averages are the stable isotope values? Averaged over the 1 m snow cores?

– Table 3: How should one read this table? Since the table is not symmetric around the diagonal, I understand that the correlation numbers differentiate between the two regions, but it is unclear which region belongs to which column or row. Also, why are there no correlation values listed for d-excess, although you mention in the text and show in Fig. 4 that there is at least some correlation between 'd' and temperature for the cDML transect data?

**References**

Ekaykin, A., Eberlein, L., Lipenkov, V., Popov, S., Scheinert, M., Schröder, L. and Turkeev, A.: Non-climatic signal in ice core records: lessons from Antarctic megadunes, The Cryosphere, **10** (3), 1217–1227, DOI: `10.5194/tc-10-1217-2016`, 2016.

Münch, T., Kipfstuhl, S., Freitag, J., Meyer, H. and Laepple, T.: Regional climate signal vs. local noise: a two-dimensional view of water isotopes in Antarctic firn at Kohnen Station, Dronning Maud Land, Clim. Past, **12** (7), 1565–1581, DOI: `10.5194/cp-12-1565-2016`, 2016.

Münch, T., Kipfstuhl, S., Freitag, J., Meyer, H. and Laepple, T.: Constraints on post-depositional isotope modifications in East Antarctic firn from analysing temporal changes of isotope profiles, The Cryosphere, **11** (5), 2175–2188, DOI: `10.5194/tc-11-2175-2017`, 2017.

---

## Author Comment (AC1) · 1 Jul 2020

**Reviewer #1**

This manuscript presents results about superficial snow cores across the Dronning Maud Land and Princess Elizabeth Land, as well as a superficial ice cores from near the DML transect. The produced data are of high value as they can help identify in two coastal areas of East Antarctica if the deposition leading to the formation of the firn is homogenous over areas characterised with largely changing conditions (temperature, accumulation…). This type of study is very valuable because they will help in the long run to improve the interpretation of the water isotopic composition from ice core records. At this stage though, I cannot recommend that the manuscript is accepted for publication. While I'm really keen on seeing these results published, I believe that the current version of the manuscript present significant flaws which do not honour the high quality of the dataset which was I'm sure produced at high costs (at least 3 Antarctic campaigns).

The authors would like to thank the reviewer for their valuable suggestions to improve this manuscript. As noted by the reviewer, this data is the result of two Antarctic campaigns. We are currently improving the manuscript with significant improvements in the introduction and the discussion sections as suggested by the reviewers. We have addressed (in blue) all the reviewer comments to improve the overall quality of the manuscript to fulfil The Cryosphere standards.

I have included some more specific points as major and minor comments on the manuscript. In general, I do not believe that the manuscript is providing any critical interpretation of the results. While the authors compared their results to various other results from past publications (for instance the diffusion models of (Münch, Kipfstuhl, Freitag, Meyer, Laepple, 2017) or the isotope-isotopes or isotope-temperature slopes from (Masson-Delmotte et al., 2008)), they fail to include the added values obtained by the comparison, or to bring up new interpretation as a scientific paper should do. In the current state, the manuscript provides and makes public this very valuable dataset, but, in my opinion, is not using it to learn something that could help the scientific community. It saddens me to give this harsh review to this manuscript which I would have liked to see through. I sincerely hope that my comments below will help the authors in the complex interpretation of this very valuable dataset.

In the revised manuscript, we focus on the spatial variability of snow accumulation and the diffusion process specifically. We also include the forward diffusion modelling of the snow core records in order to verify the diffusion process in the ice core. Additionally, the discussion on the differential diffusion and temperature reconstruction are also being included. The moisture source reconstruction using HYSPLIT back trajectories have been improved in order to add value to this manuscript.

Major Comments:

The introduction could be improved. It is difficult to follow what is the goal of the introduction in respect with the abstract, and the title of the article. At this stage, I could expect anything from a review on water isotopes in Antarctica to the interpretation of a new ice core record. Overall, the manuscript seems to cover a wide range of different aspect of what could be done with these data, but do not actually follow through on any of these. I would suggest to refocus the introduction, and the manuscript on the main point of the manuscript, which in my opinion

is the comparison between the two transects and the comparison between the ice core and the cDML transect.

We have rewritten the Introduction section with the focus on the spatial variability of stable isotopes and post depositional diffusion on ice cores with more recent relevant citations. And we have refocused on the discussion, following the reviewer's comments.

One of the major aspects comes from the evaluation of the accumulation rates for the difference sites. If I understood well the paper, they were estimated from the successive maxima/minima from the isotopic profiles, yet, there is evidence that diffusion of white noise would also create successive cycles that look similar to the seasonal cycle (Laepple et al., 2018). I think it would be very valuable for the manuscript to have rigorous tests that the accumulation rates determined here are not artefacts created by the diffusion length. Typically, a good test would be to diffuse with noise for each site and compare the mean distance between maxima/minima of the diffuse core with the one of the observed core. If the accumulation rate predicted for the diffused white noise is close to the observed accumulation, this would cast doubts on the determination of the accumulation rate in my opinion. Typically, we have been able to observe this phenomenon for accumulation rates up to 80 kg.m$^2$.a$^1$ (w.e.), and have never tried this for accumulation as high as observed here. Yet, this could also be affecting the results here for sites such as cDML 15, 24, 25. Observing the results from Fig. 5 where the diffusion length is roughly 6cm for the first meters (lines 110 to 111), one would expect white noise diffused cycles of 5 times the diffusion length, so roughly 30cm, which seems rather similar to what is observed around 4m deep in Fig. 5. Also, are the accumulation rates in water equivalent? As this will have a strong impact on the diffusion.

In order to verify the seasonal maxima/minima of the isotope profiles, we have also used major ion profiles and matched with the stable isotope profiles. Such detailed comparison would be provided in the supplementary data and clearly support that the observed seasonality is not due to artefacts. Also, as suggested by the reviewer, the amplitude of the snow core isotope profiles was diffused with white noise for each site. Results from the coastal cores show that the amplitude of the isotope profiles in the snow cores were indeed the actual amplitude and not noise and therefore would be able to support the authenticity of the accumulation rates. Such exercise will also be done for the rest of the transect and these results would be provided in the revised manuscript. The accumulation rate is given in water equivalent and shall be clarified in the revised manuscript.

The interpretation of the multiple regression model seems to be arbitrary to me in the present form. In Table 3 one can see the correlation between the different variables (even though the part of the table for PEL seems to be missing which prevents from interpreting Table 4). In cDML, roughly all the variables are very well correlated with one another, as a result, when you do the multiple regression model in Table 4, you're not proving that the distance and the elevation explain negligible variance, but that the variance explained by the temperature and the accumulation is most likely linked already to the distance and the elevation. The argument for the entire paragraph from line 135 to 145 seems completely specious to me.

Table 3 consists of correlations between different variables from both cDML and PEL which is separated diagonally along the value 1. Since this form of table is not clear to the reader, we provide a new table separately showing cDML and PEL values. We agree to the reviewer's comment on the likely link of stable isotope variability to distance and elevation. A separate

regression analysis using only the distance and elevation parameters explained a similar variance on stable isotope variability similar to temperature and accumulation in both cDML and PEL. We have changed this section of the discussion in the revised manuscript.

The comparison of the ice core and the snow cores is probably the most important aspect that could be included in this manuscript. At this stage, it is not possible to evaluate how much the signal got dampened considering how different appear to be the sites of the cDML9 and the ice core drilling site. The accumulation of the ice core drilling site appears to be twice as large as of cDML9, while this is not mentioned anywhere in the manuscript. Overall, "I have the feeling" that the ice core drilling site would match what the authors refer to as a "coastal" type of site, while it's compared to cDML9 which is classified as "mountainous". First, I should not have to guess or "have a feeling", and this should be studied in the manuscript. In general, considering that you have 25 snow cores, I feel it would make sense to align all of them to the core and to have statistical constrains on the original amplitude, as the sites were actually different (25km apart I would guess from Fig. 1).

We have compared all the cores from cDML transect in order to examine the amplitude of the seasonal stable isotope signals as suggested by the reviewer. The amplitude from all regions showed similar results which showed that the ice core record diffused over the period of five years. We include this discussion in the revised manuscript and will also try to incorporate in Fig. 5 or as a supplementary figure.

Second, there is no clear constrain of post-depositional processes here, not any use of the comparison between the snow core and the ice core. For instance, can you use this method to actually deconvolute the diffusion from the climatic signal and reconstruct the temperature from the firn core? If so, does it compare with the model temperature time series for this site? Only if you are actually able to do this have you properly constrained the post depositional processes. Also, for a coastal site like the one of the ice core, I have trouble to imagine that diffusion is the most important post-depositional process affecting your ice core, compared to sublimation/condensation combined with katabatic winds, wind redistribution or scouring…

We will use the differential diffusion signal in order to understand the differences in the diffusion of the stable isotopes which would be useful in reconstructing the temperature of the firn core record. Though diffusion is not the most critical parameter compared to wind redistribution and sublimation in the coastal, it would be an important exercise to perform in order to understand the dynamics.

Minor Comments:

Lines 58 to 60: "After recovery, the snow cores were transferred directly into precleaned high-density polyethylene bags and sealed immediately to avoid contamination during storage and shipped under frozen conditions to the National Centre for Polar and Ocean Research, India. The cores were stored at –20°C till analysis." Is there any evidence that diffusion cannot happen in highly porous snow at -20°C? These conditions are very close to summer firn conditions at places like Dome C, where diffusion still takes place.

Since the snow/ice cores were collected from coastal sites where summer temperatures could touch around 0°C, storage under -20°C conditions are sufficient to ensure the protection from diffusion. In order to further check this, we used fresh sub-samples from the archived snow cores and analyzed these samples using the laser based Triple Isotope Water Analyzer. Results

from these analyses and comparison with older analyses using IRMS showed that there was no diffusion in the amplitude of the snow cores which have been well preserved in sealed extra polypropylene boxes at -20°C.

Lines 75 to 77: "The seasonality in snow cores was determined by establishing the summer and winter peaks in isotope records where a minimum amplitude of 4‰ between summer and winter was used to differentiate these peaks as detailed in figure 2 our previous publications (Mahalinganathan et al., 2012; Mahalinganathan and Thamban, 2016)." There are evidences that the cycles observed in isotopic profiles of water isotopes can be linked to diffused white noise, especially for low accumulation areas (Laepple et al., 2018), as a result, they can be deceived when used to identify annual layers and date ice cores. It is not necessarily going to be the case for snow cores that close to the coast, but it's definitely something good to discuss.

We agree that the diffused white noise impact the isotope profiles in the low accumulation areas. However, the present transects showed an accumulation rate that are over 100 kgm$^{-2}$yr$^{-1}$ even in the inland sections of the cDML region. We have analyzed the possibilities of white noise diffusion and will be discussed in the revised manuscript.

Lines 88 to 93: Are the accumulation rates obtained from the distance between maxima/minima in the isotopic profiles? In which case, as mentioned in my previous comment, I'd strongly suggest to provide evidence that similar distance would not be created artificially by diffusion of white noise.

As described earlier, we have added major ion profiles and matched with the stable isotope profiles to verify the seasonal demarcations. Also, as suggested by the reviewer, the amplitude of the snow core isotope profiles is being diffused with white noise for each site.

Lines 112 to 113: "Five-day back-trajectory frequency maps of coastal, mountainous and inland locations showed vast differences in the sources between summer and winter (Fig. 6)" Are they vastly different? I'm a bit confused for several reasons. First, what months did you choose for summer and winter? Antarctic summer is rather brief for a lot of sites, with a large asymmetry between summer months (DJF), and winter months (AMJJASO) (Van Den Broeke, 1998). As a result, statistically, you could already explain having less events in summer, and as a result, a small range of possible storms visible in the sample. Second, it seems that if the dark red and dark green envelopes do cover a large area, most of the trajectory originates from very similar cones both in summer and in winter, which is quite unexpected, and very intriguing. Considering that the dark red corresponds most likely to a very small number of trajectory, it would be interesting to have quantitative numbers of trajectories for each of the sectors you mention here, to actually be able to evaluate information beyond the plot. Finally, this does not necessarily reflect the actual contribution to the different ice cores, indeed, a small number of events can contribute a large amount of the accumulation, and I have serious doubts that the trajectories originating from the Plateau contribute for a lot of the accumulation for instance. I would recommend to realise an analysis similar to the one found in Figure 5 of (Genthon, Six, Scarchilli, Ciardini, Frezzotti, 2015) and include the relative contributions to the snow fall amount for clusters of trajectories.

We used DJF for summer and JJA for winter, representing the peak summer and winter months. While Antarctica has extended periods of winter, most studies consider JJA as the core winter months. The 6 hourly trajectories for each day extending back to five days for 3 months each

(summer / winter) were made from the representative locations and all these trajectory data was used to make this trajectory frequency map. We do agree that the accumulation in Antarctica is mainly produced by a few large precipitation events. The trajectories attenuate at 5 days and those trajectories that arrive from plateau does not indicate the origin of moisture source. In order to give more clarity, we shall quantify the trajectory clusters (similar to Genthon, 2015) instead of the frequency map, as suggested by the reviewer.

Lines 119 to 120: "Spatial variations of snow accumulation in Antarctica are primarily due to the presence of physical barriers during snowfall and snow redistribution post deposition (Melvold et al., 1998; Vaughan et al., 1999)." I would say I disagree with this sentence, and that there are a lot of literature that has been produced since 1999 going in a completely different direction. Depending on your definition of "spatial variations of snow accumulation", you can obviously interpret this sentence in a lot of different ways, so if you look at small scales, it's been shown that at Dome C, the accumulation can vary by large amount over short distance without any physical barriers like mountains (Genthon et al., 2015). Looking at typical coast-to-interior patterns, I would recommend for instance to consider (Agosta et al., 2019) which shows that even for other sites without mountain ranges, the dominant pattern is the coast-to-plateau gradient.

We shall include the recent references suggested by the reviewer and discuss the possibilities of factors other than the physical barriers influencing the spatial variability of snow accumulation in this section.

Lines 125 to 130: "These mountain chains in cDML act as a physical barrier to the air masses arriving from the Southern Ocean impacting the snow accumulation and redistribution. As a result, the study area could be separated into three distinct accumulation regimes. The physiography and topography of the cDML region evidently influenced the snow accumulation rates showing a strong correlation with distance and elevation (Table 3). On the contrary, the PEL transect showed moderately high accumulation with little variation between the coastal ($276 \text{ kgm}^2 \text{ yr}^1$) and the inland ($260 \text{ kgm}^2 \text{ yr}^1$) sections." While it's true that it seems that the accumulation gradient is larger for the DML site, and that one could expect the mountain range to affect the accumulation along the transect, I don't think you've proven it yet. Indeed, the PEL transect does not reach altitude as high as the cDML transect, which could also explain the difference in accumulation. As previously mentioned, I would also be careful with the evaluation of the accumulation rates from the isotopes, considering you're using the temperature from RACMO, I would suggest to also include the accumulation rates from RACMO which might help in the interpretation here (similar to (Agosta et al., 2019)), or to even remove this paragraph.

We have now included RACMO model accumulation rates and will be discussed as suggested.

Lines 137 to 138: "However, the multiple regression models using the geographical parameters and 18O showed negligible variance with distance and elevation (Table 4)." Considering how well correlated are the temperature, the elevation and the distance to the coast, I don't think you can make this assumption here.

We tested the elevation and distance parameters with the stable isotope variability separately and we agree with the reviewer. We have modified the discussion.

Lines 150 to 151: "The slope of the LMWL in cDML (7.9) is lower than that of the global meteoric water line (GMWL) while the slope of LMWL in PEL (8.12) had a slope close to GMWL." It would help here to have error bars on the MWL slope, as well as correlation coefficient and significance tests to assess the robustness of the variations of the slopes of 0.1 around the MWL.

The error bars are included as suggested.

Lines 163: "Therefore, the proposed spatial slope (0.80‰ /C) by Masson-Delmotte et al. (2008) seems to be reasonable." I would say that the slopes you obtain seem reasonable compared to the ones found in (Masson-Delmotte et al., 2008), considering that in the aforementioned article were included over 1000 snow pits and ice cores, across all over Antarctica. The two sentences are equivalent as the statement is a bijection, but it seems more that your results are validated by what was already found in this study, than you are validating this previous study considering the content provided in both cases.

We agree with the reviewer's comment and we have changed this sentence.

Lines 180 to 181: "The detailed stable isotope records and chronology of this ice core is discussed in an upcoming paper (Tariq et al., 2020, unpublished)." Considering that the synchronisation of both cores is key, while the Tariq et al is not available to evaluate how the two records was synchronised, it is difficult to evaluate the work in the section. Indeed, the accumulation for cDML9 is 157 kg.m$^{-2}$.a$^{-1}$, assuming that the value is in water equivalent (which it ought to be), this means that in 5 years, you expect 2.75m of accumulation, very far from what is shown in Fig. 5. As the core is still quite far from the site, it is possible that the accumulation was slightly different, and the value of cDML9 seems to be much lower than the neighbouring sites, so even taking into account the values for cDML8 and 10, we would expect 3.94m, quite short to what is described in Fig 5. If the accumulation rates at the ice core site are that different from the accumulation rates along the transect, can you provide evidence that the amplitude of seasonal cycle of isotopic composition in the precipitation was the same? Typically, considering that the cDML transect sites neighbouring the ice cores are all mountainous sites, while the ice core seems to be in a more "coastal" sites, can you illustrate if you obtain similar amplitudes in the different firn cores of the transect that would justify that the original amplitude of the ice core could reasonably be close to what was in cDML9. Considering the large difference of amplitude between what is found for cDML9 (>5permil) and the surface of the ice core in summer/winter 2012-2013 (<2permil), one could also wonder if the ice core site just has singularly less pronounced seasonal cycle of precipitation isotopic composition.

We are most certain that the ice core record near the mountainous section is diffused over the time period of five years. This is clear when we compare all the snow cores from cDML transect (except that of cDML 1 and cDML 14 which showed no clear signals) which showed higher seasonal amplitude. These profiles shall be made available in the revised manuscript, either in the form of a supplement or in the main text with examples from representative sections of the transect.

Figure 5: Which snow core is included in the figure? I couldn't find the information easily.

The snow core mentioned here is cDML 9. We will now also include cores from coastal section and this information shall be included in the revised manuscript.

---

## Author Comment (AC2) · 1 Jul 2020

**Reviewer #2**

The authors present new stable isotope and accumulation data retrieved from two spatial (coast to inland) transects in the East Antarctic regions of central Dronning Maud Land (cDML) and Princess Elizabeth Land (PEL). The data were obtained from drilling and analysing short (1 m) snow profiles at 25 and 21 positions along the respective transects, covering coastal, mountainous (cDML) and plateau regions. Based on the data, the authors present standard analyses of the relationships between oxygen and hydrogen isotopic data as well as between isotopic data and local temperature and accumulation rate, and how these vary between the two transects. In addition, they present brief diffusion and back-trajectory analyses. While the paper presents new data, the overall quality and scientific originality of this work, as I will outline in my general comments below, does not meet the standards of The Cryosphere. Therefore, I rate this manuscript as being not acceptable for final publication in this journal.

We are grateful to the reviewer for providing their constructive comments. We have addressed (in blue) all the reviewer comments to improve the overall quality of this manuscript to fulfill The Cryosphere standards.

**General Comments:**

The major shortcomings of this work are its low quality and scientific originality. Regarding quality, the writing suffers from many small grammatical errors, which make the manuscript hard to read. Additionally, the introduction is poorly structured, the methods incomplete, and the results read like a dry technical document using many repetitive phrases. Overall, I would strongly suggest the authors to consult a language editing service to improve grammar and style. In addition, there seem to be some inconsistencies between the data as given in the tables and presented in the figures (see my specific comments).

The quality of the manuscript is being improved significantly with changes in the Introduction section, including missing details in the methodology section and undertaking more data analysis and discussion as suggested by both reviewers. The inconsistency in data and figures / tables are also addressed. English language improvement would be made as suggested in order to improve the quality of the manuscript.

Furthermore, the paper lacks significant scientific originality. While the authors present an extensive data set from two expensive transect sampling campaigns, which in principle would offer the chance for an interesting study, they unfortunately fail to exploit this scientific potential. What the authors present is a set of analyses which are standard for isotope studies and which already have been conducted and shown many times before. This is especially sad since the authors seem to be well aware of the challenges and uncertainties related to the interpretation of (surface snow) isotopic data in Antarctica as it has been developed by the recent literature. However, by contrast, this study does not offer any new insights which could aid the community in gaining a deeper understanding of the involved processes shaping firn and ice isotopic records. The two following examples are symptomatic for this deficiency. The authors make use of state-of-the-art firn diffusion modelling to estimate the amount of diffusion which should have attenuated an isotopic profile over the course of time, but they miss to actually test this diffusion model for their chosen site by quantitatively comparing how the model prediction fits to the data. The second example is the discussion of the back-trajectory analysis. The most text of the respective discussion section

actually presents additional analysis results that were not mentioned in the results section before, while the small portion of interpretation remains at a poorly speculative level, which leaves the reader guessing what we can learn from this analysis. An interesting aspect of the presented data set is the different importance of the spatial accumulation variability for the spatial isotope variability of the two transects. This would be an interesting candidate for an in-depth study, which the authors however do not pursue. Also, the transect data seems to comprise at least 1m long isotope profiles for more than 40 positions, but the authors do not analyse or show any of these profiles, instead sticking to the analysis of only the mean values. Overall, the manuscript fails to meet the scientific criteria of The Cryosphere and is not in a state to be accepted for final publication.

Scientifically, we have now included the profiles of isotopes from the representative regions and will provide the complete depth profiles of isotopes as a supplement and discuss the major features in the text. We have included the discussion on how the diffusion model fits to the actual ice core data as suggested by the reviewer. We have also included the details of the back-trajectory analysis in the results section and improved the discussion on the same in order to understand the possible moisture sources for the study region. We also attempt to improve the discussion on the spatial variability of accumulation and stable isotopes in the two transects as noted by the reviewer. We are confident that these measures undertaken would substantially enhance the quality of the manuscript to the standards of The Cryosphere.

**Specific comments:**

Below I list some specific comments that could help to improve the manuscript, but I do not include the many occurrences of grammatical errors, missing specific articles, etc. L1: The abstract should briefly introduce the background of the study in a first sentence.

The background of the study is included in the beginning of the abstract as suggested.

L1: Please consider using the correct terminology "stable water isotopologues", or use phrases such as "stable isotopes" or "isotopic composition".

The correct terminology, "stable isotopes", is being used throughout in the revised manuscript.

L3: Please introduce $\delta^{18}O$ and $\delta^2H$ shortly here, or alternatively, rewrite this passage and introduce the notation in the introduction.

We have introduced the $\delta^{18}O$ and $\delta^2H$ in the abstract as suggested.

L8: The term HYSPLIT is not explained anywhere here. Please also note in this regard that you do not explain how the back-trajectory analysis is actually conducted, whether in the Methods nor the results text.

HYSPLIT exercise along with the details of back-trajectories are now explained in the methodology section of the revised manuscript.

L18: Why is high resolution important in this context?

The high accumulation in the coastal Antarctic region permit us to have snow cores representing detailed changes that occurred over a complete year. As a result, we have over 40 snow core depth

profiles representing the spatial changes over central Dronning Maud Land and Princess Elizabeth Land, with a sampling resolution in terms of months (or large precipitation events), which typically lacks in short ice cores. The combination of areal coverage with high temporal resolution aids in the interpretation of regional temperature and climate while giving insights on the ice core records.

L25: I do not see any obvious direct connection between the Town et al. study on atmosphere-surface snow exchange and wind-driven erosion and redistribution processes. You have to elaborate more on this, and consider adding more relevant recent literature, e.g. the stratigraphic noise studies by Münch et al. (2016, 2017) and the mega dune studies by Ekaykin et al. (e.g. Ekaykin et al., 2016). L27; sentence "The deuterium excess [...]": There is no logical link to the previous text. L27; sentence "Studies by...": There is no logical link to the previous text. L36; "no net change [...] were observed": This statement is misleading. For normal firn diffusion and densification, we do not expect any net mean change, as a basic property of these processes. As it is written, one could interpret this as something which just has not been observed yet due to measurement uncertainty or such. L36: The statement that diffusion in ice is negligible has to be put into context. Ice diffusion is certainly not negligible for very deep ice cores = very old ice! L36; "and therefore gets preserved": Unclear from the context, what gets preserved? L43 first sentence: This statement is in direct contradiction to the sentence about the studies by Steen-Larsen et al. (2014) and Ritter et al. (2016) a few sentences above. LL15-50: Overall, the introduction is poorly structured with missing logical links between sentences and paragraphs. You may consider rewriting it altogether.

The introduction section is now rewritten with more focus, clarity and citing more recent relevant literature while maintaining logical flow as suggested in the specific comments L15-50. Statements on firn diffusion and densification with respect to interpretation has been clarified in the revised manuscript.

L67: What do you mean by external precision? To which isotope species do you refer here – $\delta^{18}O$ or $\delta^2H$? Why do you measure samples also on the Los Gatos device, if all samples have been already analyzed on a mass spectrometer (cf. L65 " All samples...")?

Conventionally, the external precision on stable isotopes of water are estimated based on the repeated analysis of international standards like Standard Mean Oceanic Water (SMOW). Since such standards are limited and precious, all water isotope measurements are made with reference to a laboratory standard calibrated with SMOW. In current case, the precision determined by repeatedly measuring a laboratory standard called Central Dronning Maud Land snow (cDML) is used for estimating external precision. Here we run the cDML standard 20 times – where this standard is passed through the autosampler and all the lines that a sample would be going through, and arrive in the precision value. All samples were first analysed using IRMS. We freshly sub-sampled and some of these samples were reanalysed with the newly acquired LGR Triple Isotope Water Analyzer that uses Off-Axis Integrated Cavity Output Spectroscopy (OA-ICOS) technique to measure $\delta^{18}O$ or $\delta^2H$ after 9 years. Since data generated through this method also revealed similar amplitude, we demonstrate that there exist no diffusion in stable isotopes while storage. Such completely different methods used in this study confirms the reliability of the stable isotope data presented here.

LL75-76: As you also have impurity data available from the same cores (as evident from previous publications), it would be probably more robust to conduct the dating using isotope and impurity species together, instead of setting an arbitrary threshold on the isotope peak heights.

We have already used the ionic concentrations along with stable isotopes in order to date the snow cores as cited in L77. However, since this is not properly conveyed in the manuscript, representative examples of the isotope profiles along with the impurity species will be included in the revised manuscript.

LL51-86: Descriptions of the HYSPLIT back-trajectory analysis and the used multiple regression model is missing.

The HYSPLIT description and details of the multiple regression model are now included in the revised manuscript.

L95: It would be actually nice to see some example isotope profiles or even use them to further study the spatial variability and its drivers along the two transects.

Isotope profiles from representative sections of topography will be included in the revised manuscript.

LL107-109: Using the data provided in Tables 1 and 2, I find for both transects a significant correlation between d-excess and temperature on the 0.1 significance level (R = -0.4, p = 0.06 for cDML; R = 0.39, p = 0.08 for PEL) – can you comment on this? In any case, concerning the opposite correlations and the significant scatter of the data (Fig. 4), I however ask myself what we can actually learn from regressing 'd' against temperature here and if there is actually any meaningful explanation behind the correlations?

All the parameters except the d exhibited a strong correlation between each other at a higher significance level (0.01) in both cDML and PEL. In the revised manuscript, we include the d-excess temperature relationship in Table 3, and also include a more-detailed discussion on this relationship as suggested.

L137: What kind of multiple regression model do you use here? This could be added to the Methods section.

We use the multiple linear regression model and we clarify this in the revised manuscript.

L153: What do you mean here with "all the samples"? Did you use the combination of cDML and PEL samples for another $\delta^{18}O$ - $\delta^2H$ regression?

We intended to convey that we used all the sub-samples rather than mean values at every core location in cDML and PEL transects. This is clarified in the revised manuscript.

L163; "seems to be reasonable": This is expressed the wrong way around and anyway a quite obvious result: Your slope results from two Antarctic subset regions scatter around the Antarctic wide mean slope – as it is to be expected.

We agree. We have now revised and compare our work with the compilation of Antarctic wide dataset here and we have changed this sentence.

L169-172: It should be mentioned that the relationship by contrast is positive for the PEL transect data at the same significance level (according to my estimate, see the previous comment above), so a more detailed discussion of the d-excess-to-temperature relationship is needed here.

A more detailed discussion explaining the data from the PEL transect shall be included in this section as suggested by the reviewer.

L188-189: What exactly are the initial and final depths here?

This equation was intended to calculate the differential diffusion signal between $\delta^{18}O$ and $\delta^2H$ in order to reconstruct the temperature, however was not utilized in the manuscript. In the revised manuscript, we will include a discussion on the temperature reconstruction from the firn core as suggested by the reviewer 1.

L189: Münch et al. (2016) is not the correct reference; I guess you mean Münch et al. (2017), where a similar approach was taken.

Correct citation is provided in the revised manuscript.

Section 4.2: What do we learn from this? Have you tried to forward-diffuse the younger record with the estimated diffusion length to see if the mismatch in seasonal amplitude is really only a result of diffusion? By which value did you shift the newer data downwards and how did you choose this value? If you really want to constrain the post-depositional changes at this site similar to the study by Münch et al. (2017), you also have to take into account the effects of densification and stratigraphic noise, but I also think that the extent of overlap your two records have is much too small to arrive at any meaningful conclusions.

In this section, we aim to understand the process of diffusion over time, by comparing the snow core record with a firn core that was drilled 5 years after the snow core. The exercise was a direct comparison of the amplitude diffusion between the snow core and the firn core. In order to improve the discussion here, we shall include the forward-diffusion of the snow core record to test the diffusion length. Though the record may be small, the higher resolution of the snow core record would help us understand the diffusion process in the high accumulation region.

Section 4.3: This section effectively presents additional results from the back-trajectory analysis, which should be placed therefore into the respective results section. Beyond that, it is pretty much unclear to me what we learn from this exercise.

In order to provide more clarity, we shall quantify the trajectory clusters instead of the frequency map, as suggested by the reviewer. We also have improved the discussion in this section to understand the origin of moisture sources.

Figure 1: Please explain the abbreviations ETOPO1, IBSCO and RAMP2, and provide a source for these map data.

The abbreviations and map sources are provided in the revised manuscript.

Figure 4: While I can reproduce the cDML plot given the data in Table 1, the PEL plot using the data of Table 2 looks different than the one provided here. Please check the consistency of all your data and figures; see also my comment on Fig. 3.

We apologize for this error. The temperatures provided in Table 2 was of a different model, while the figure used the RACMO temperature. This inconsistency in table 1 will be fixed in the revised manuscript.

Table 1; caption: What kind of averages are the stable isotope values? Averaged over the 1m snow cores?

The averages mentioned in Table 1 are annual average values and not 1 meter average. After the demarcation of a year, the values over a year are averaged.

Table 3: How should one read this table? Since the table is not symmetric around the diagonal, I understand that the correlation numbers differentiate between the two regions, but it is unclear which region belongs to which column or row. Also, why are there no correlation values listed for d-excess, although you mention in the text and show in Fig. 4 that there is at least some correlation between 'd' and temperature for the cDML transect data?

Table 3 intends to showcase the correlation values from both cDML and PEL regions – which are split diagonally along the value 1. Since this form of table is complicated, we provide a new table separately showing cDML and PEL values. All values in this table was at 0.01 significance level while the cDML correlation showed 0.05 significance level. These values would be added in the revised manuscript with clear marking.